# A model for designing intraocular pressure-regulating glaucoma implants

Inês C. F. Pereira[1,2], Hans M. Wyss[1,2], Leonard Pinchuk[3,4], Henny J. M. Beckers[5], Jaap M. J. den Toonder[1,2]*

1 Microsystems, Department of Mechanical Engineering, Eindhoven University of Technology, Eindhoven, The Netherlands, 2 Institute for Complex Molecular Systems (ICMS), Eindhoven University of Technology, Eindhoven, The Netherlands, 3 InnFocus, Inc., a Santen Company, Miami, Florida, United States of America, 4 Ophthalmic Biophysics Center, Bascom Palmer Eye Institute, University of Miami Miller School of Medicine, Miami, Florida, United States of America, 5 University Eye Clinic Maastricht, Maastricht University Medical Centre+ (MUMC+), Maastricht, The Netherlands

* j.m.j.d.toonder@tue.nl

**Data Availability Statement:** All relevant data are available within the manuscript and its Supporting information file.

**Funding:** This research was financially supported by Chemelot Institute for Science & Technology

## Abstract

Glaucoma is a group of eye conditions that damage the optic nerve, the health of which is vital for vision. The key risk factor for the development and progression of this disease is increased intraocular pressure (IOP). Implantable glaucoma drainage devices have been developed to divert aqueous humor from the glaucomatous eye as a means of reducing IOP. The artificial drainage pathway created by these devices drives the fluid into a filtering bleb. The long-term success of filtration surgery is dictated by the proper functioning of the bleb and overlying Tenon's and conjunctival tissue. To better understand the influence of the health condition of these tissues on IOP, we have developed a mathematical model of fluid production in the eye, its removal from the anterior chamber by a particular glaucoma implant–the PRESERFLO® MicroShunt–, drainage into the bleb and absorption by the sub-conjunctival vasculature. The mathematical model was numerically solved by commercial FEM package COMSOL. Our numerical results of IOP for different postoperative conditions are consistent with the available evidence on IOP outcomes after the implantation of this device. To obtain insight into the adjustments in the implant's hydrodynamic resistance that are required for IOP control when hypotony or bleb scarring due to tissue fibrosis take place, we have simulated the flow through a microshunt with an adjustable lumen diameter. Our findings show that increasing the hydrodynamic resistance of the microshunt by reducing the lumen diameter, can effectively help to prevent hypotony. However, decreasing the hydrodynamic resistance of the implant will not sufficiently decrease the IOP to acceptable levels when the bleb is encapsulated due to tissue fibrosis. Therefore, to effectively reduce IOP, the adjustable glaucoma implant should be combined with a means of reducing fibrosis. The results reported herein may provide guidelines to support the design of future glaucoma implants with adjustable hydrodynamic resistances.

(InSciTe) under grant agreement BM3.03 SEAMS. The funders had no role in study design, data collection and analysis, decision to publish, or preparation of the manuscript.

## Introduction

Glaucoma is the second leading cause of preventable blindness worldwide, with over 100 million people expected to suffer from the disease by 2040 [1]. Elevated intraocular pressure (IOP, above 21 mmHg) remains the most important known risk factor for the development and progression of glaucoma. IOP is determined by the balance between the production of aqueous humor within the eye and its drainage out of it through two distinct pathways–the trabecular and the non-trabecular pathway. In patients with primary open-angle glaucoma, there is an abnormal increase of resistance to aqueous outflow through the trabecular outflow pathway, which leads to a build-up of fluid in the eye that results in high IOP [2]. Hence, current treatment options aim to lower the IOP with the goal of preventing additional glaucomatous optic nerve damage [3]. The application of topical ocular hypotensive drug agents is often chosen as the first-line treatment, but fundamental challenges to pharmacological therapy continue to exist, including local and systemic adverse effects and poor patient adherence [3, 4]. Laser-therapy is considered when the visual field continues to deteriorate despite maximum use of topical medication, and if unsuccessful, incisional surgery is considered. Conventional filtration surgeries include trabeculectomy and implantation of glaucoma drainage devices, and both procedures are based upon the same principle: creating an alternative drainage route that allows the aqueous humor to escape from the anterior chamber of the eye as a means of lowering IOP [5]. Conventional aqueous shunts and some of the new minimally-invasive glaucoma surgery (MIGS) devices [6], drain the aqueous humor via a shunt into the subconjunctival/ sub-Tenon's space, where a fluid reservoir known as filtering bleb is formed, as shown in Fig 1. The aqueous humor can then traverse the bleb into the surrounding conjunctival tissue and can be absorbed by the subconjunctival or episcleral vasculature [7].

The bleb and the overlying conjunctiva are considered to be the cornerstone of IOP control following glaucoma filtration surgery [8]. The favorable outcome of filtration surgery is dictated by the proper functioning of the bleb, which is highly dependent upon postoperative modifications in the conjunctival tissue resulting from the healing response [9]. Conjunctival

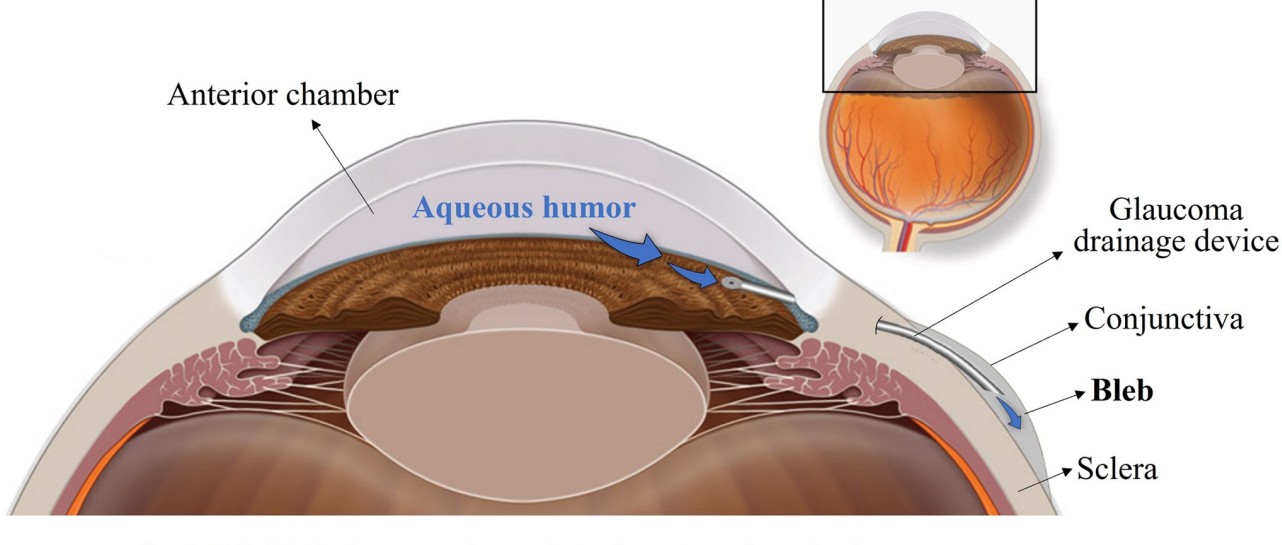

**Fig 1. Aqueous humor artificial drainage pathway.** Schematic representation of the artificial drainage pathway of aqueous humor into the subconjunctival space, where a filtering bleb is formed, following glaucoma drainage device insertion.

wound healing is a complex multifactorial process consisting of a cascade of overlapping events, including hemostasis, inflammation, cell proliferation and tissue remodeling [10]. During this process, there is a high propensity of subconjunctival fibrous tissue formation, i.e. scarring, resulting in bleb encapsulation (formation of a bleb capsule), which may increase the resistance to aqueous outflow through the bleb wall, thus reducing subconjunctival absorption of aqueous humor. This leads to an elevation of the IOP to preoperative or even higher values, and consequently failure of the glaucoma filtration surgery [7]. There are various mechanisms postulated to contribute to subconjunctival scarring. These mechanisms include previous surgical procedures breaching the conjunctiva, a long history of topical medication, predisposition to conjunctival inflammation, aqueous humor composition, subconjunctival flow rate and direction, as well as hydrostatic pressure acting on the bleb [11]. Antifibrotic agents such as mitomycin C and 5-fluorouracil have been increasingly used to control the wound healing process and increase surgical success, however they are often associated with serious complications such as bleb leak and hypotony [12, 13]. Besides subconjunctival scarring that leads to increased IOP, hypotony is one of the most common complications following glaucoma surgery. It is defined as very low IOP of 5 mmHg or less, and it may lead to vision loss in up to 20% of the patients who experience hypotony [14]. The acute inflammatory response that naturally follows incisional surgery may also contribute to hypotony–more permeable/leaky subconjunctival capillaries may absorb the aqueous humor at a faster rate than it is produced, thus leading to over-filtration [15].

Although better designed glaucoma implants have emerged in recent years, made from superior materials that evoke minimal tissue inflammation, excessive fibrosis with scar tissue formation and hypotony-related complications are still frequently reported. One reason behind this is that such devices are totally passive, i.e., the drainage of aqueous humor depends upon a fixed hydrodynamic resistance of the shunt [16]. In many cases, however, the hydrodynamic resistance of the shunt may not be optimal which may lead to high IOP, when the resistance in the subconjunctival space is too high (due to the presence of a scar layer), or to over-drainage, if the resistance is too low (resulting in hypotony) [17]. To determine the ideal hydrodynamic resistance that a glaucoma implant must have to overcome these two most common postoperative complications–bleb scarring and hypotony–we developed a model that calculates the pressure in the bleb under these conditions. A porous media model was used to model aqueous humor flow through the bleb and subconjunctival tissue, and its absorption by the subconjunctival vasculature. The model accounts for the bleb size and shape, hydraulic conductivity of the subconjunctival tissue, as well as its fluid absorptive capacity, among other parameters. According to the calculated bleb pressure, the implant design and dimensions, and consequently its hydrodynamic resistance was tuned to achieve a healthy IOP of approximately 10 mmHg.

## Methods

The mathematical model used for the calculation of bleb pressure, describing the fluid removal from the anterior chamber of the eye through a glaucoma drainage device, its drainage into the filtering bleb and absorption by the subconjunctival tissue, is based on the work of Gardiner and co-workers [15]. Using this model, Gardiner and co-workers investigated how the IOP is influenced by several factors, including the aqueous humor production and outflow rates, bleb geometry, subconjunctival tissue conductivity and tissue absorptive capacity. We, on the other hand, applied an adapted version of the model to study the possibility to control intraocular pressure in glaucoma patients by using a glaucoma implant with adjustable hydrodynamic resistance. To the best of our knowledge, this is the first time that such a model was

used to obtain insight into the adjustments in the implant's hydrodynamic resistance that are required for IOP control when the two most common postoperative complications following glaucoma filtration surgery take place, hypotony and bleb scarring.

The commercial finite element method (FEM) package COMSOL Multiphysics was used to numerically solve the mathematical model. The default solver settings were used, and a total of 767,912 triangular mesh elements were created. The maximum element size was set to 0.005 mm and local mesh refinements were not needed, since the average quality of the mesh was 95.98%, and the minimum quality obtained was 55.59%.

## Geometry

The model is comprised of two domains: one representing the bleb and subconjunctival tissue, and the other representing the glaucoma drainage device. A tube-like shunt with the same length and lumen diameter as the PRESERFLO® MicroShunt (Santen, Osaka, Japan) was used to represent the glaucoma drainage device. Its design and dimensions are shown in Fig 2 [18]. The sclera was not included in our model. Although it is reported in the literature that some aqueous humor flowing into the bleb after glaucoma filtration surgery is reabsorbed by the episcleral vasculature, tracer-based studies have proved that the conjunctival blood vessels and conjunctival lymphatic system are the main structures involved in removing the excess interstitial subconjunctival fluid (aqueous humor) [8, 19–21]. Taking this into account, we have decided to not include the absorption of aqueous humor by the episcleral vasculature in our model and only consider the conjunctival route of absorption.

The extent and elevation of a filtering bleb vary widely among patients and are important indicators of bleb function. The dimensions of a healthy bleb and overlying conjunctiva/Tenon's capsule are shown in Fig 3A and are based on an Optical Coherence Tomography (OCT) image of a functioning bleb, in a patient with a healthy IOP of 10 mmHg. These dimensions

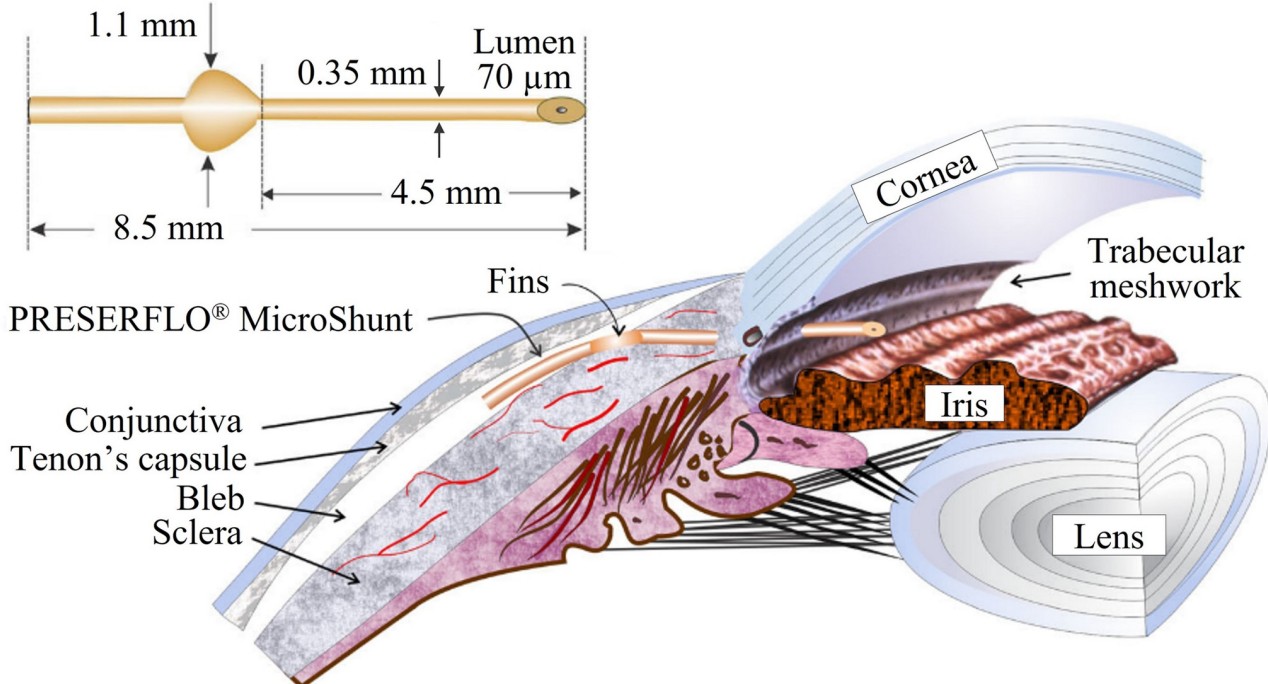

**Fig 2. The PRESERFLO® MicroShunt.** Schematics showing the PRESERFLO® Microshunt dimensions and placement in the eye [18].

**Fig 3. Two-dimensional geometry of the filtering bleb.** (**A**) Shape and dimensions of a healthy bleb and overlying conjunctival/Tenon's tissue based on a cross-sectional Optical Coherence Tomography (OCT) image (patient with an IOP of 10 mmHg); these dimensions were used to simulate the normal (healthy/well-functioning bleb) and hypotony cases. (**B**) Shape and dimensions of the bleb, scar tissue layer, and subconjunctival tissue used to simulate the bleb scarring scenario.

were used to simulate the normal case-scenario (healthy/well-functioning bleb) and hypotony cases. To simulate the fibrotic case we have chosen a lower bleb height, as it is commonly reported that scar tissue formation leads to progressive flattening of the bleb, and in some cases even to its disappearance [8]. Additionally, for this scenario a new domain representing the scar layer was added at the interface between the bleb and the subconjunctival tissue, as shown in Fig 3B. The thickness of the scar tissue is dependent upon the natural healing response of each individual after the glaucoma filtration surgery and may vary over time. Nevertheless, in our simulations we have used a constant thickness and only varied the hydraulic conductivity of the tissue to evaluate its impact on the bleb pressure and IOP. It is also generally reported that more fibrosis occurs directly above the center of the bleb, which is where the aqueous humor exits the glaucoma drainage device [7]. Therefore, the thickness of the scar layer used in this model is 100 μm larger at the center than on the lateral sides.

We assume that the bleb is axisymmetric. Thus, a 2D-axisymmetric model is adopted in which we consider a circular region of the subconjunctival tissue centered on the bleb, as depicted in Fig 4. Although not shown, a layer of scar tissue optionally covers the surface of the bleb, depending upon the case study to simulate, as explained above (see Fig 3).

## Governing equations

The aqueous humor is mainly composed by water (99%), so we describe its behavior by the well-known incompressible Navier-Stokes equation in the shunt domain, coupled with Darcy's law, describing flow in porous media, in the bleb/subconjunctiva domain [22]. Neglecting the gravitational acceleration and assuming steady state, the Navier-Stokes and mass conservation equations describing the fluid flow in the shunt are given as [23]

$$\nabla \cdot \mathbf{v} = 0 \tag{1}$$

$$\rho(\mathbf{v} \cdot \nabla \mathbf{v}) = -\nabla p + \mu \nabla^2 \mathbf{v}, \tag{2}$$

Where $\rho$[kg m$^{-3}$] and $\mu$[Pa s] are the density and dynamic viscosity of aqueous humor, respectively, $\mathbf{v}$[m s$^{-1}$] is the velocity vector, and $\rho$[Pa] is the pressure. For the boundary conditions, a mass flow rate calculated from a volumetric flow rate ($Q_{in}$) of 2.5 μL min$^{-1}$ was assumed at the inlet of the shunt. The value of 2.5 μL min$^{-1}$ was chosen as an approximated average value of aqueous humor production rate during a period of 24 hours. The choice of this value was supported from the literature [24]. For simplification, we did not consider the outflow of aqueous humor through the natural outflow pathways (trabecular and non-trabecular/uveoscleral) in

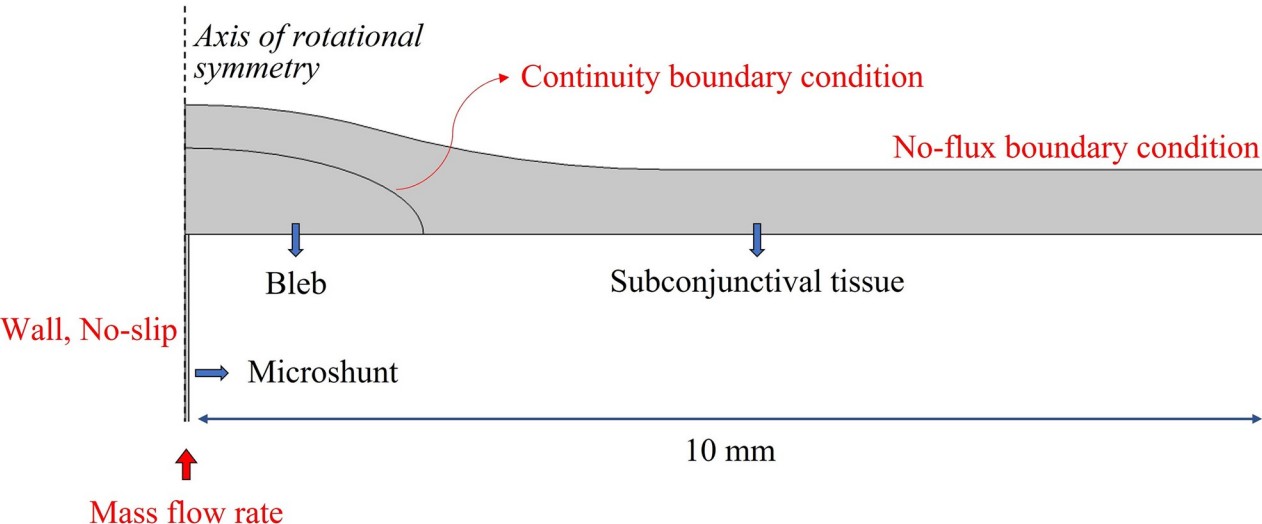

**Fig 4. 2D-axisymmetric computational domain of the subconjunctival drainage of aqueous humor through a hollow tube-like microshunt.** The bleb dimensions are as in Fig 3 and a scar tissue layer may be included as in Fig 3B. The applied boundary conditions are indicated in red.

our simulations like done by Gardiner and co-workers [15]. The reason for this is that, in a glaucomatous eye and when a glaucoma drainage device is implanted, most aqueous humor drains through the glaucoma implant into the bleb, which represents the easiest fluidic pathway [19]. It is hypothesized in the literature that only 10% of the aqueous humor drains through the natural outflow pathways after trabeculectomy is performed [20].

The average pressure in the bleb $p_{bleb}$ calculated using Darcy's law is applied on the outflow surface of the shunt. Since the wall of the shunt is impermeable, it was modelled with stationary rigid boundaries at which a no-slip boundary condition was imposed.

To describe the movement of fluid into the bleb, through the subconjunctival tissue and its removal by the microvasculature, Darcy's law for flow in porous media was used. Besides the subconjunctival tissue, we considered the bleb as a porous medium since it has numerous microcysts (clear spaces) filled with aqueous humor, lined by epithelium containing goblet cells, and surrounded by a superficial stroma that is composed of loosely arranged connective tissue of irregular collagen with a slight or absent subepithelial inflammatory response of lymphocytes, macrophages, and fibroblasts [11, 21, 25–29]. To couple the equations for the fluid in the shunt domain with Darcy's equation in the bleb, we applied the normal inflow velocity of the fluid leaving the tube as the inlet boundary condition of the bleb, and as mentioned above, we have attributed the average pressure calculated in the bleb as the outlet boundary condition of the tube.

Darcy's empirical observations demonstrated that the fluid velocity in porous media is proportional to the pressure gradient; therefore, fluid transport in the porous media can be described by [30]

$$\mathbf{v}_i = -K\nabla p_i, \qquad (3)$$

Where $\mathbf{v}_i$[m s$^{-1}$] is the interstitial fluid velocity (often referred to as the Darcy velocity), $K$[m$^2$ s$^{-1}$ Pa$^{-1}$] is the hydraulic conductivity of the tissue, and $p_i$[Pa] is the hydrostatic pressure in the tissue interstitium. The hydraulic conductivity is the proportionality constant in Darcy's law, and we note here that its value will vary spatially in our model to represent the properties of

different tissues, such as scar tissue and subconjunctiva. Additionally, the hydraulic conductivity may vary over time, as a result of the healing response following surgery [15].

The mass balance equation for a steady state incompressible fluid shows that the divergence of the velocity is zero,

$$\nabla \cdot \mathbf{v_i} = 0. \tag{4}$$

This equation is adequate for porous media when there is no fluid source or sink in the medium. However, in biological tissues fluid is exchanged between the interstitial space and the capillaries, which in our model acts as a sink for the fluid. Thus, the steady state incompressible form of the continuity equation becomes

$$\nabla \cdot \mathbf{v_i} = \phi_v, \tag{5}$$

Where $\phi_v$ is the rate of fluid flow per unit volume from the vasculature into the interstitial space, or vice versa, and can be evaluated through Starling's law as [31, 32]

$$\phi_v = L_P \frac{S_A}{V} (p_v - p_i - \sigma[\pi_v - \pi_i]), \tag{6}$$

Where $L_p[\text{m}^2 \text{ s}^{-1} \text{ Pa}^{-1}]$ is the hydraulic permeability of the blood vessel wall, $\frac{S_A}{V}[\text{m}^{-1}]$ is the surface area of blood vessel walls per volume of tissue for transport in the interstitium, $p_v[\text{Pa}]$ is the microvasculature pressure, $\pi_v$ and $\pi_i$ [Pa] are the plasma and interstitial fluid oncotic pressures (the osmotic pressure induced by plasma proteins), respectively, and $\sigma$ is the average reflection coefficient for the plasma proteins (a measure of protein permeability). A reflection coefficient equal to 1 means that the vessel wall is impermeable to plasma proteins, whereas a coefficient equal to 0 means that there is no transport restriction [33]. Because most capillaries in the body are fairly impermeable to high molecular weight proteins, and based on literature research, we have decided to use a reflection coefficient of 0.91 in our simulations. The values of the remaining parameters used in the numerical simulations are listed in Table 1, along with references to the experimental studies they are based on.

We assume that the bleb does not contain vessels, which means that there is no fluid exchange in this domain. Hence, the values attributed to the variables of Starling's equation

**Table 1. Parameter values used in the simulations.**

| Parameter | Assumed value | | Reference |
|---|---|---|---|
| $\mu$ dynamic viscosity of aqueous humor | $7{,}5 \times 10^{-4}$ Pa s | | [23] |
| $\rho$ density of aqueous humor | $998{,}7$ kg m$^{-3}$ | | [23] |
| $L_P$ hydraulic permeability of blood vessel wall | Hypotony | $1 \times 10^{-8}$ m s$^{-1}$ Pa$^{-1}$ | [34] |
| | Normal and Bleb scarring | $1 \times 10^{-10}$ m s$^{-1}$ Pa$^{-1}$ | |
| $\frac{S_A}{V}$ vessel wall area per tissue volume | $6{,}7 \times 10^3$ m$^{-1}$ | | [35] |
| $p_v$ vasculature pressure | $1{,}3 \times 10^3$ Pa | | [15] |
| $\pi_v$ vessel oncotic pressure | $2{,}6 \times 10^3$ Pa | | [15, 36] |
| $\pi_i$ interstitium oncotic pressure | $1{,}3 \text{xx} 10^3$ Pa | | [15, 36] |
| $\sigma$ reflection coefficient | $0{,}91$ | | [37] |
| $K$ hydraulic conductivity | Bleb | $1{,}5 \times 10^{-8}$ m$^2$ s$^{-1}$ Pa$^{-1}$ | [15, 38–40] |
| | Subconjunctival tissue | $1{,}5 \times 10^{-11}$ m$^2$ s$^{-1}$ Pa$^{-1}$ | |
| | Scar tissue | $3 \times 10^{-13}$ m$^2$ s$^{-1}$ Pa$^{-1}$ | |
| | | $2 \times 10^{-13}$ m$^2$ s$^{-1}$ Pa$^{-1}$ | |
| | | $1 \times 10^{-13}$ m$^2$ s$^{-1}$ Pa$^{-1}$ | |

**Table 2. Boundary conditions used in the simulations for the shunt and bleb/scar layer/subconjunctival tissue domains.**

| Domain | Surface | Boundary conditions | Mathematical expression |
|---|---|---|---|
| **Tube** | Inlet | Mass flow rate | $\dot{m} = \rho \cdot Q_{in}$, <br> $\rho$ is the aqueous humor density; <br> $Q_{in}$ is the volume flow rate (2.5 µL/min). |
| | Outlet | Pressure | $p_{out} = p_i = p_{bleb}$, <br> $p_{bleb}$ is the average pressure calculated in the bleb domain. |
| | Side walls | No-slip | $\mathbf{v} = 0$, <br> $\mathbf{v}$ is the velocity vector. |
| **Bleb/Subconjunctival tissue** | Inlet | Normal inflow velocity (from the tube domain) | $\mathbf{v} = -nu_0$, <br> n is the boundary normal pointing out of the domain; $u_0$ is the normal inflow speed, calculated as follows $u_0 = \mathbf{n} \cdot (\mathbf{v}_r + \mathbf{v}_z)$, $\mathbf{v}_r$ and $\mathbf{v}_z$ being the velocity components in the r and z directions, respectively. |
| | Side walls | No flux | $-\mathbf{n} \cdot \rho_{\mathbf{v}} = 0$ |
| | Interface bleb/scar layer/ subconjunctival tissue | Continuity | $\mathbf{v}_{bleb} = \mathbf{v}_{—scar\ layer} = \mathbf{v}_{subconjunctival\ tissue}$ <br> $p_{bleb} = p_{scar\ layer} = p_{subconjunctival\ tissue}$ |

are only applicable to the subconjunctival tissue. Additionally, due to the extensive hydration of the bleb we assume that it offers little resistance to fluid flow, and therefore we give it a hydraulic conductivity that is 1000 times higher than that of the subconjunctiva. To investigate the influence of the extent of fibrosis on the aqueous filtration capacity and, consequently, on the IOP, the hydraulic conductivity of the scar tissue layer was varied between $1 \times 10^{-13} - 3 \times 10^{-13} \text{ m}^2 \text{ s}^{-1} \text{ Pa}^{-1}$ 1. Regarding the hydraulic permeability of the blood vessel wall in the subconjunctival tissue, we have chosen a lower value to simulate the hypotony case. We have decided to use this value, which is generally reported for tumorous tissues, because in the early post-operative period the tissue is undergoing swelling and acute inflammatory response, which typically results in increased vasculature permeability (leaky vessels) [32, 37, 41].

For the boundary conditions in the bleb/subconjunctiva domain, a no-flux boundary condition was applied on all external boundaries, including the surface in contact with the sclera and except for a central circle at the bottom center of the bleb where the microshunt is positioned. A continuity of flux and pressure was assumed at the interface between the bleb and the subconjunctiva (or between the bleb, scar tissue, and subconjunctiva for the bleb scarring scenario).

The boundary conditions used in the simulations for the shunt and bleb/scar layer/subconjunctival tissue domains are shown in Table 2.

## Model validation based on literature evidence

The validation of the developed model makes use of the available evidence on IOP outcomes after the implantation of the PRESERFLO MicroShunt. Cases of hypotony are rare with this device, especially due to its small lumen diameter of 70 µm [18, 42]. However, there are still patients suffering from this vision-threatening condition. Similarly, and although the material that this implant is made of–poly(styrene-*block*-isobutylene-*block*-styrene), or "SIBS"–is proven to be highly inert and biocompatible, bleb encapsulation cases are still frequently reported [43, 44]. These conditions are associated with extreme, unhealthy IOPs. We compared these values typically reported in clinical trials with those calculated in our simulations to validate the model.

## Adjustable glaucoma implant

To obtain insight into the adjustments in the implant's hydrodynamic resistance that are required for IOP control, we have simulated the flow through a microshunt with an adjustable

lumen diameter (adjustable hydrodynamic resistance) and its drainage into the subconjunctival space for the different scenarios. In the hypotony case, the simulation was performed with lumen diameters varying between 40 and 70 μm, and for the bleb scarring scenario the lumen diameter was changed from 70 to 370 μm. The hydrodynamic resistance of the microshunt with different lumen diameters was determined, and the resultant IOP was calculated. The hydrodynamic resistance was calculated as

$$r(\text{mmHg}/\mu\text{L min}^{-1}) = \frac{IOP - p_{\text{bleb}}}{Q_{\text{in}}}, \tag{7}$$

where the IOP is the pressure calculated at the inlet of the glaucoma drainage device, $p_{\text{bleb}}$ is the pressure calculated in the bleb, and $Q_{\text{in}}$ is the flow rate (2.5 μL min$^{-1}$). Ideally, an adjustable glaucoma implant must be able to switch between fluidic resistances in order to control and maintain the IOP within healthy values (5–15 mmHg), irrespective of the condition of the bleb/subconjunctival tissue.

## Experimental validation

The calculated IOP for the two case scenarios studied with varying shunt lumen diameters was validated experimentally by performing microfluidic tests. For this, two different microchips were fabricated: one for the hypotony model validation, and the other for the encapsulated model validation. In each device, four square cross-section channels were implemented, as shown in Fig 5.

All channels have the same length as the PRESERFLO MicroShunt, i.e., 8.5 mm. The height and width of the channels in the hypotony device are, respectively: 37x37, 46x46, 55x55 and 64x64 μm, and in the encapsulated bleb device these are: 64x64, 155x155, 247x247 and 338x338 μm. These dimensions correspond to different circular tube diameters as can be seen in Table 3.

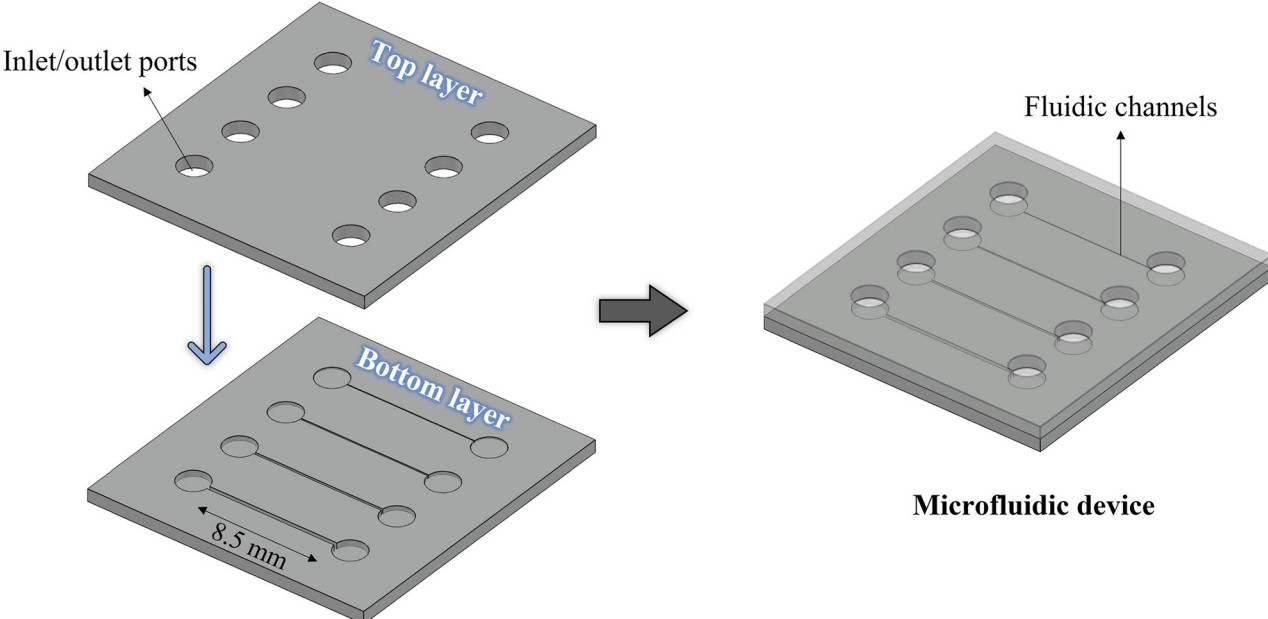

**Fig 5. Design of the microfluidic devices.** Schematic illustration of the design of the microfluidic devices used for the model validation (right side). The bottom and top layers the microdevice is made of are shown on the left side.

**Table 3. Dimension of the channels of the microfluidic chips and corresponding tube diameters.**

| Case scenario | Channel dimensions (height x width) | Corresponding circular tube diameter |
|---|---|---|
| Hypotony | 37 x 37 μm | 40 μm |
| | 46 x 46 μm | 50 μm |
| | 55 x 55 μm | 60 μm |
| Hypotony and Encapsulated bleb | 64 x 64 μm | 70 μm |
| Encapsulated bleb | 155 x 155 μm | 170 μm |
| | 247 x 247 μm | 270 μm |
| | 338 x 338 μm | 370 μm |

The chosen tube diameters are within the range of diameters also used in the simulations (see previous section "Adjustable glaucoma implant"). To determine the height and width ($h$) of a square channel with the same hydraulic resistance ($r$) of a circular channel, the following formula was applied:

$$r_{(\text{square channel})} = r_{(\text{circular channel})} \tag{8}$$

$$\frac{12\ \mu L}{1 - 0.917 \times 0.63}\ \frac{1}{h^4} = \frac{8}{\pi}\ \mu L\ \frac{1}{a^4}, \tag{9}$$

Where $\mu$[Pa s] is the dynamic viscosity of the fluid, $L$ is the length of the channel/tube, $h$ is the height/width of the square channel, and $a$ is the radius of the circular tube.

The microfluidic chips were made from SIBS, the same highly bioinert and biocompatible material that the PRESERFLO MicroShunt is made of. The SIBS pellets with a 30% styrene content were generously provided by InnFocus Inc., a Santen Company (Santen, Osaka, Japan). The devices were fabricated by replica molding with hot embossing, using femtosecond laser machined fused silica glass molds. Femtosecond laser-assisted chemical wet etching has been investigated as an alternative process to fabricate micro-devices [45]. It is based on a two-step process of ultrashort-pulsed laser radiation in transparent materials, followed by chemical wet etching to selectively remove the exposed material. The laser beam, focused inside the glass, locally modifies its refractive index and chemical properties, and creates patterns that can be used to, by chemical etching, generate three-dimensional structures with high precision, aspect ratio and complexity [45, 46]. Using this technique, different channel heights can be easily fabricated within one microfluidic chip. This would be extremely cumbersome using other classical manufacturing techniques, such as photolithography, due to the need of multiple photolithography steps with precise alignment, which is very difficult to achieve even when using a mask aligner [47].

The design of the mold for the microfluidic chips was prepared using the dedicated Alpha-cam software, where the laser scanning path (tool-path) to be fed to the FEMTOprinter f200 aHead (FEMTOprint SA, Switzerland) for exposing the fused silica glass, is also generated. The mold was fabricated on 75x25x1 mm fused silica glass slides. The pulse energy and repetition rate used were 230 nJ and 1000 kHz, respectively. The laser was focused with a Thorlabs 20X microscope objective with a numerical aperture (NA) of 0.4. When the machining program was finished, the glass slide was immersed in a concentrated solution of 45% potassium hydroxide (KOH, Sigma-Aldrich) diluted in water to remove the exposed material. Finally, the mold was rinsed thoroughly with acetone and de-ionized (DI) water to remove all debris. To facilitate the release (demolding) of the patterned SIBS after the hot embossing step (described next), the femtosecond laser-machined glass mold was first coated with a superhydrophobic layer of fluorosilane (Trichloro(1H,1H,2H,2H-perfluorooctyl)silane, Sigma-Aldrich). To

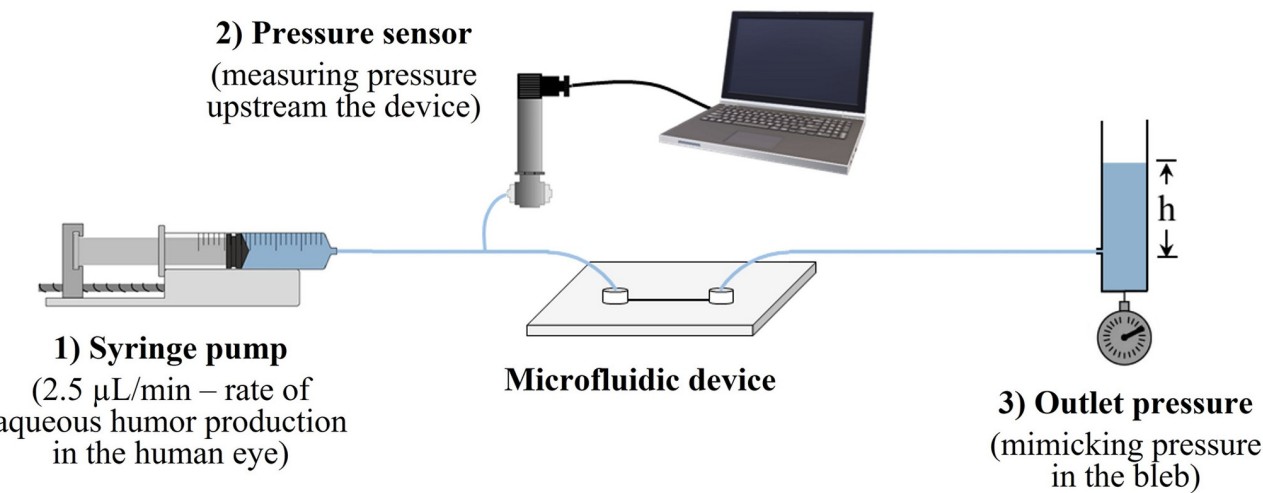

**Fig 6. Setup used for the microfluidic experiments.**

improve the adhesion of this coating, the mold underwent an oxygen plasma treatment performed immediately before the fluorosilane vapor deposition. After the silanization treatment, the mold was ready to be used in the hot embossing machine (Specac limited) together with SIBS pellets to fabricate the microfluidic chips. Hot embossing is a micro-fabrication technique in which micron-scale structures present in a mold are replicated on to a polymer substrate by application of pressure and temperature. We used 150°C to melt the SIBS and 5 tons of pressure to transfer the features in the mold to the SIBS film. The demolding took place after the hot embossing had cooled down to 80°C. The patterned SIBS film was then cut into a smaller 16.5x15 mm rectangular film, with the channels centered in the middle, forming the bottom layer of the microfluidic device. The top layer of the device was made of another SIBS rectangular film, but without any imprinted features, and containing the inlet and outlet connections. A biopsy punch was used to create the connection holes. To obtain a closed microfluidic device, bottom and top layers were thermally bonded on a hot plate at 90°C for 10 min, while applying pressure with a weight placed on top of the device.

The setup used for the microfluidic experiments is illustrated in Fig 6 and was comprised of: 1) a syringe pump (Fusion 200, Chemyx Inc.), pumping DI water at a constant flow rate of 2.5 μL min$^{-1}$ –rate of aqueous humor production in the eye; 2) a pressure transducer (Omega Engineering), connected to the syringe pump and to the inlet of the device, thus constantly measuring the pressure upstream the device (in mmHg); and 3) a column of water connected to the outlet of the device, mimicking the pressure in the bleb.

The height of the column of water was adjusted to match the calculated bleb pressures for the different situations studied. The pressure measured at the inlet of the device would correspond to the IOP in a real case scenario. This pressure was measured in four separate samples for each type of microfluidic chip. Then, the measured pressure was finally compared with the calculated IOPs for the different bleb conditions with varying shunt lumen diameters.

## Results and discussion

In this section, numerical results obtained for bleb pressure in each of three simulated postoperative conditions–hypotony case, healthy/well-functioning bleb, and encapsulated bleb–are presented. Next, the IOP values calculated when the PRESERFLO MicroShunt is used as the glaucoma drainage device, are compared with the IOP values that are generally reported in the

literature following implantation of this device. Finally, we explore the different hydrodynamic resistances that a future adjustable, patient-specific glaucoma implant needs to be able to cover to maintain a healthy IOP. The IOPs calculated using our model are compared to pressures experimentally measured upstream of microfluidic devices that contain channels with different dimensions (thus, different hydrodynamic resistances) when applying constant flow rate while setting the outlet pressure, as a basic *in vitro* model of the implanted glaucoma drainage device.

## Bleb pressure and IOP

Fig 7 shows the model predictions of the interstitial pressure in the subconjunctival space for three different scenarios: (1) when the subconjunctival space is inflamed in the early period after surgery, which is likely to result in hypotony; (2) in the presence of a healthy or well/functioning bleb; and (3) when excessive fibrosis leads to bleb encapsulation. First, it is possible to verify that for all the investigated situations the interstitial pressure reaches its maximum at the center of the bleb, directly above the distal end of the glaucoma drainage device. The mechanical stresses imposed by the aqueous humor outflow at this location may lead to more vigorous scarring, which explains the commonly observed thicker capsule at this central area. This also suggests that the scar tissue forms first above the bleb, and then grows until covering it completely [7, 9, 48]. In implants where a mechanism to restrict flow is applied in an early period after surgery, a lower incidence of bleb encapsulation is reported on the long-term [49, 50]. This delayed aqueous outflow allows for a maturated bleb to form, which can then more efficiently resist to elevated hydrostatic pressures.

The graph in Fig 8 shows the values of interstitial bleb pressure and IOP calculated for each of the investigated case-scenarios. For a well-functioning bleb and when the PRESERFLO MicroShunt is used as the glaucoma drainage device, the calculated IOP is 10.42 mmHg, which is consistent with the values generally reported in the literature [42]. The presence of the scar tissue encapsulating the bleb leads to a higher bleb pressure, and consequently to an increase in the IOP, as can be inferred from Fig 8. The fibrotic tissue is a vascularized, entangled network of collagen fibers, which along with deposition of glycosaminoglycans and

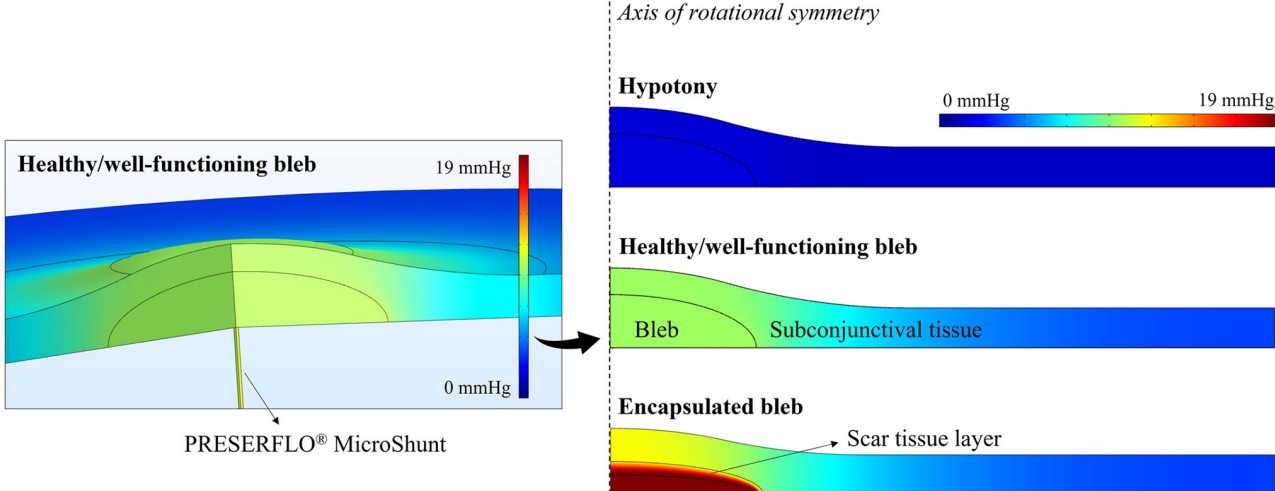

**Fig 7. Interstitial pressure distribution in the subconjunctival space.** Model prediction of the interstitial pressure distribution in the subconjunctival space in the presence of hypotony, healthy/well-functioning bleb, and encapsulated bleb (scar layer with hydraulic conductivity of $2\times10^{-13}$ m$^2$ s$^{-1}$ Pa$^-$). In all cases, the PRESERFLO MicroShunt is used as the glaucoma drainage device. Color scale indicates interstitial fluid pressure in mmHg.

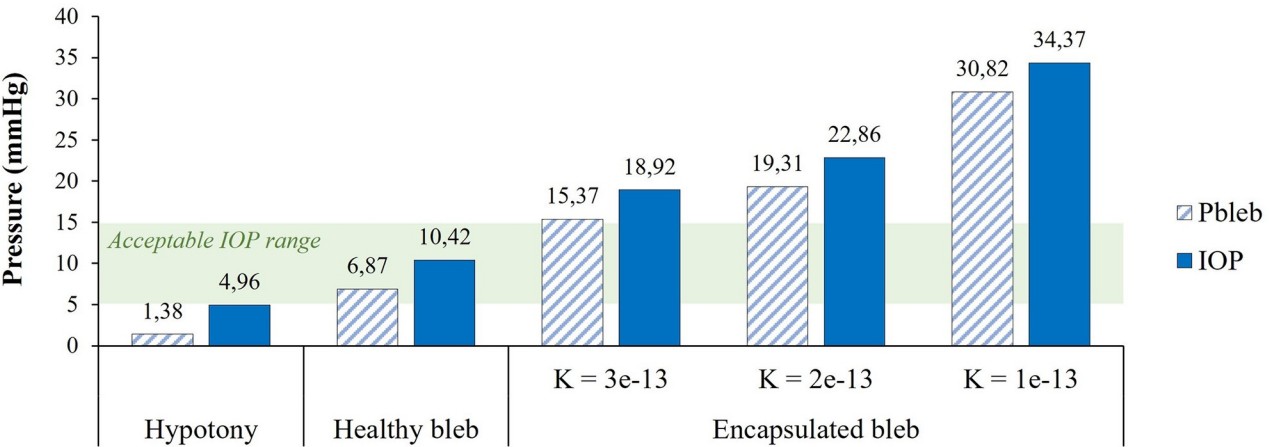

**Fig 8. Bleb pressure ($p_{bleb}$) and IOP calculated for each of the case-scenarios studied: Hypotony, healthy bleb, and encapsulated bleb.** For the latter case, the hydraulic conductivity of the scar layer varies from $1 - 3 \times 10^{-13}$ m$^2$ s$^{-1}$ Pa$^{-1}$. In all cases, the PRESERFLO MicroShunt is used as the glaucoma drainage device. The shaded green area represents an acceptable IOP range of 5–15 mmHg.

proteoglycan core proteins are associated with low tissue hydraulic conductivity [40]. This hydraulic conductivity is dependent on the extent of the fibrotic reaction. The more severe the fibrotic response is, the lower is the hydraulic conductivity of the scar tissue and the higher will be the IOP. A decrease in the hydraulic conductivity effectively blocks the flow of aqueous into the subconjunctival tissue, thus decreasing its absorption by the microvasculature. This leads to the accumulation of fluid in the anterior chamber which results in a high IOP. In an extreme case, fluid will not be able to flow out of the bleb anymore and the bleb is no longer an alternative outflow pathway of aqueous humor, bringing the IOP back to values of an unoperated eye. Although we have considered the presence of vasculature in the scar tissue in our simulations, which also plays a role in the absorption of fluid, this seems not to be enough to compensate for the decreased hydraulic conductivity of the tissue. This result is consistent with the uncontrolled IOP that is usually seen in thick-walled blebs, even though they appear to be well vascularized [51].

The IOP value calculated when a high permeability of the microvasculature is considered (hypotony case) is very close to the IOP upper limit used to define hypotony (5 mmHg). This is in line with the literature published on the PRESERFLO MicroShunt, where no cases of severe hypotony (IOP much lower than 5 mmHg) are reported. Only a very small percentage of patients suffer from mild transient hypotony, which usually resolves within a few days [42]. Hypotony can be aggravated if a lower rate of aqueous humor production is considered. Although we have used a constant, average value of aqueous inflow rate in our simulations, this value varies significantly between the waking and sleeping hours. In a healthy person, the aqueous humor production rate is approximately 3 μL min$^{-1}$ in the morning, 2.5 μL min$^{-1}$ in the afternoon, and 1.5 μL min$^{-1}$ during the night [52]. This varies not only with the circadian rhythm, but also among individuals. If the lower value of aqueous flow rate (1.5 μL min$^{-1}$) is considered when simulating the hypotony case, then the IOP predicted by our model decreases from 4.96 mmHg to 3.43 mmHg. This value can be even lower in the presence of bleb leakage, for example due to the application of antimetabolites (antifibrotic agents) such as Mitomycin C in the site of implantation [53, 54]. Furthermore, if we simulate over-drainage using, for instance, the 304 μm lumen diameter of the Baerveldt implant (Johnson & Johnson Vision, California, USA) and considering a normal aqueous flow rate of 2.5 μL min$^{-1}$, then the resultant IOP is even more vision-threatening– 1.39 mmHg from our model. This is the reason

why techniques to restrict early fluid flow through these big-lumen implants, or conventional aqueous shunts, were soon established, including ligating the tube externally with an absorbable ligature or placing an intraluminal suture in the tube [6, 55]. However, ligature/intraluminal suture mispositioning may occur, which then often still leads to early hypotony. Additionally, wound healing time varies widely among patients, and delayed hypotony may still occur in some cases when the temporary outflow restriction is gone [56].

## Adjustable glaucoma implant

Considering the complications associated with current (passive) glaucoma implants discussed so far, it becomes clear that an individualized surgical treatment for glaucoma patients is needed. A patient-specific glaucoma drainage device with an adjustable hydrodynamic resistance would be ideal, especially when the outflow of aqueous humor through such a device could be fine-tuned during the postoperative follow-up visits according to the IOP measured in the patient's eyes. In order to identify the hydrodynamic resistance adjustments that are necessary for achieving a healthy IOP when hypotony is most likely to occur, or when bleb scarring takes place, we varied the lumen diameter of the PRESERFLO MicroShunt to evaluate its influence in the IOP. The results are shown in Fig 9.

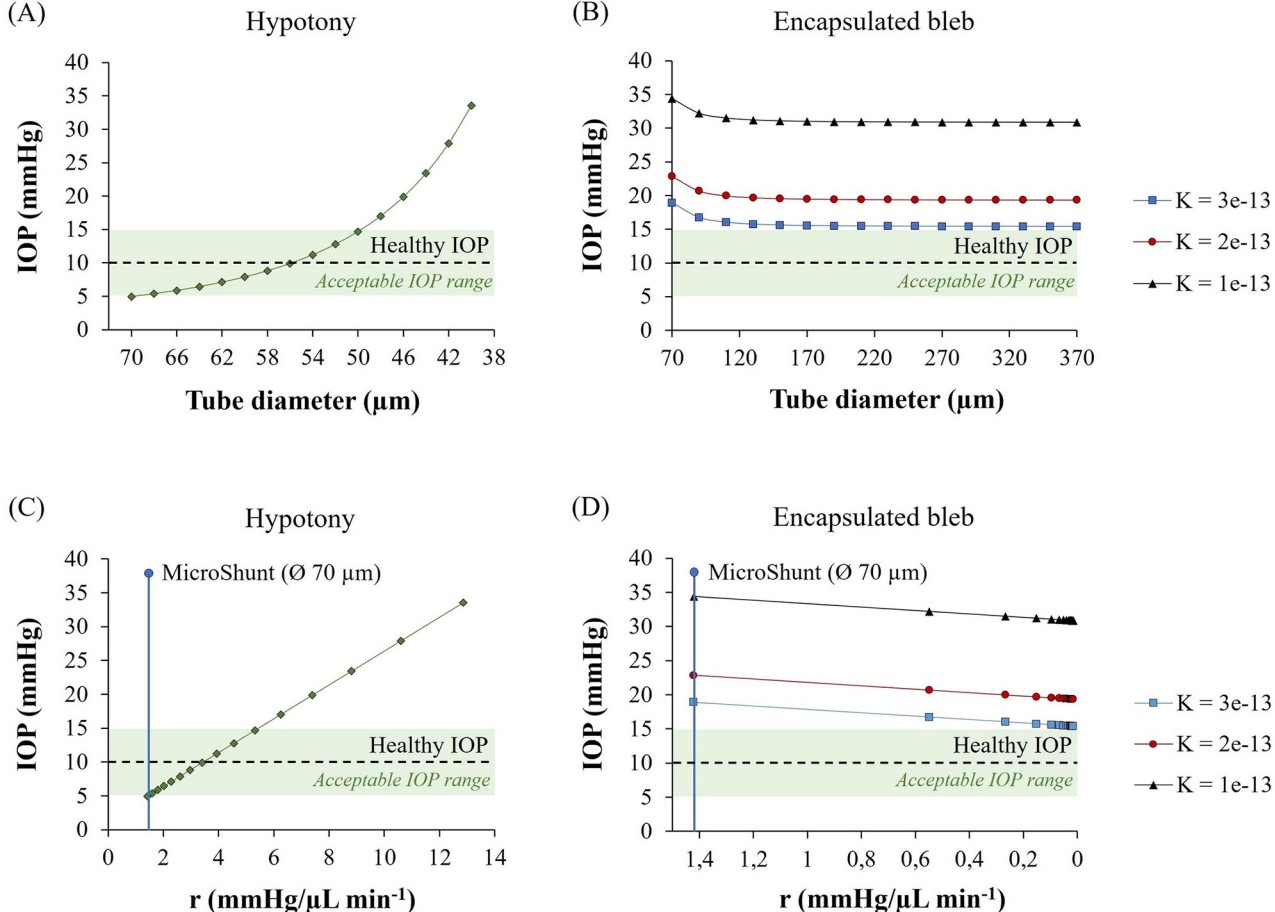

**Fig 9. Model prediction of the IOP with varying MicroShunt lumen diameters.** (**A**) and (**B**)–IOP variation with varying MicroShunt lumen diameter in case of hypotony and encapsulated bleb, respectively. (**C**) and (**D**)–Hydrodynamic resistance (*r*) of the different lumen-diameter MicroShunts and its impact on the final IOP. The shaded green area represents an acceptable IOP range of 5–15 mmHg.

As shown in Fig 9A, in case of hypotony, changing the effective lumen diameter of the tube from 70 to 55 μm raises the IOP to an acceptable, healthy value of 10 mmHg. On the other hand, in the case of high IOP due to bleb scarring as shown in Fig 9B, enlarging the lumen diameter even to very high values, does not result in the IOP decreasing sufficiently to reach acceptable values. For instance, when a scar layer with hydraulic conductivity of $3x10^{-13}$ $m^2$ $s^{-1}$ $Pa^{-1}$ is considered, the IOP of 18.92 mmHg, which occurs for a 70 μm diameter, does not become lower than 15 mmHg by enlarging the effective tube diameter even beyond 370 μm. This means that the fibrotic tissue encapsulating the bleb will always be the limiting factor, as it blocks the fluid flow into the subconjunctival tissue thus hampering its absorption by the microvasculature. Therefore, we can conclude that the adjustable glaucoma implant should be always combined with a means to reduce/limit fibrosis.

The calculated hydrodynamic resistances can be translated into different implant designs than the straightforward tube with constant circular cross-section such as the MicroShunt simulated here. Fig 9C shows that to maintain the IOP within healthy levels in the likely case of hypotony, no matter the implant design (with one or multiple channels, with micro-valves integrated, etc.), its hydrodynamic resistance needs to be at least approximately 1.4 mmHg/(μL $min^{-1}$ to avoid lowering of IOP below 5 mmHg, and it must be increased to 3.4 mmHg/μL $min^{-1}$ to level the IOP to a healthy value of 10 mmHg. For the encapsulated bleb case, the hydrodynamic resistance can also be decreased to the minimum resistance acceptable in terms of device dimensions (bigger channels will result in bigger outer device dimensions), but only a maximum IOP decrease of around 3 mmHg will be achieved, as shown in Fig 9D.

## Experimental validation results

To confirm the IOP values calculated for different bleb pressures (resulting from different bleb conditions) and for different shunt lumen diameters with distinct hydrodynamic resistances, we have performed microfluidic experiments. For this, two microchips with four channels each were fabricated. One device was used for the hypotony model validation, whilst the other was used for the encapsulated bleb model validation. Fig 10A shows a picture of the FEMTO-print glass mold used in the hot embossing machine for the fabrication of these devices. Fig 10B shows a microscopic image of a replica molded bottom SIBS device layer, and Fig 10C shows a closed, bonded device used in the microfluidic experiments.

The graphs presented in Fig 11 show a comparison between the IOPs calculated using our model and the IOPs measured in the microfluidic experiments. The simulations were performed using the geometry/dimensions of the channels present in the microfluidic devices and using the dynamic viscosity of water at 20˚C (0.001 Pa s) instead of aqueous humor, since DI water was the fluid used in the experiments. From Fig 11 we can conclude that there are no significant differences between the calculated and measured pressures, which indicates that our model correctly calculates the inlet pressure (IOP) given a certain outlet pressure (bleb pressure) and device hydrodynamic resistance. In the hypotony case (Fig 11A), it is possible to verify that a healthy IOP is achieved when decreasing the channel dimensions from 64x64 μm to 55x55 μm (equivalent to a 70 to 60 μm circular lumen diameter). For the encapsulated bleb scenario (Fig 11B–11D), we can confirm that increasing the channel dimensions from 64x64μm to 338x338μm (equivalent to 70 to 370 μm circular lumen diameter) only results in a pressure drop up to 3 mmHg. This is verified for all the hydraulic conductivities of the scar tissue layer considered. These results are in line with the simulation results presented in Fig 9.

**(A)** FEMTOprint mold

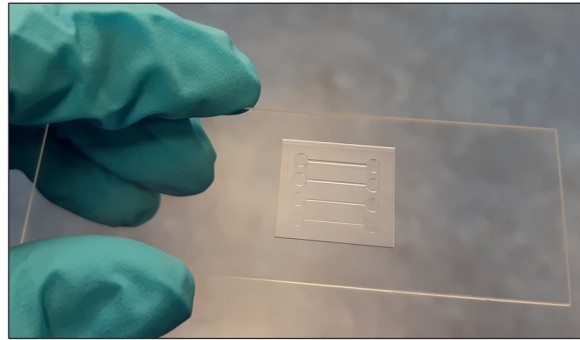

**(B)** Bottom layer of microfluidic device

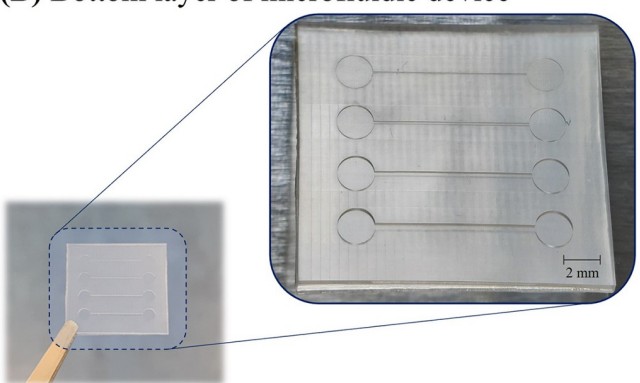

**(C)** Microfluidic device

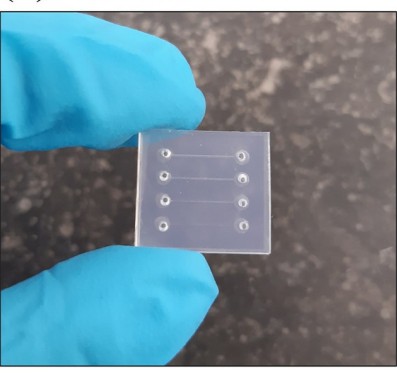

**Fig 10. Fabricated microfluidic devices. (A)** FEMTOprint glass mold used for the fabrication of the bottom layers of the microfluidic chips (in this case, for the encapsulated model validation devices). **(B)** SIBS bottom layer of the encapsulated model validation device replicated from the mold. **(C)** Final microdevice used in the microfluidic experiments.

## Conclusion

We studied the possibility to control IOP in glaucoma patients by using an adjustable implanted glaucoma drainage device, for different postoperative conditions. To this end, we developed a model describing the fluid removal from the anterior chamber of the eye through the glaucoma drainage device, its drainage into the filtering bleb and absorption by the sub-conjunctival tissue. In the model, the fluid transport in the bleb and subconjunctival tissue is simulated using Darcy's law for fluid flow inside a porous media. To account for the aqueous humor absorption by the subconjunctival vasculature, Darcy's equation is modified by employing Starling's law. The model was numerically solved using the commercial FEM package COMSOL. We simulated three postoperative conditions following glaucoma drainage device implantation–hypotony, healthy/well-functioning bleb, and encapsulated bleb due to tissue fibrosis. A tube-like shunt with the same dimensions as the PRESERFLO MicroShunt was used to represent the glaucoma drainage device in our simulations. The predicted results of IOP are consistent with the evidence available in the literature on the PRESERFLO Micro-Shunt, where only a few cases of (mild) hypotony are reported, and the IOP is reported to be approximately 10 mmHg (healthy IOP) when a well-functioning bleb is present. When bleb encapsulation takes place, the scar tissue covering the bleb hinders the flow of aqueous humor into the subconjunctival tissue where it should be absorbed by the microvasculature. As a result, the IOP increases. Our findings suggest, as expected, that the lower the hydraulic

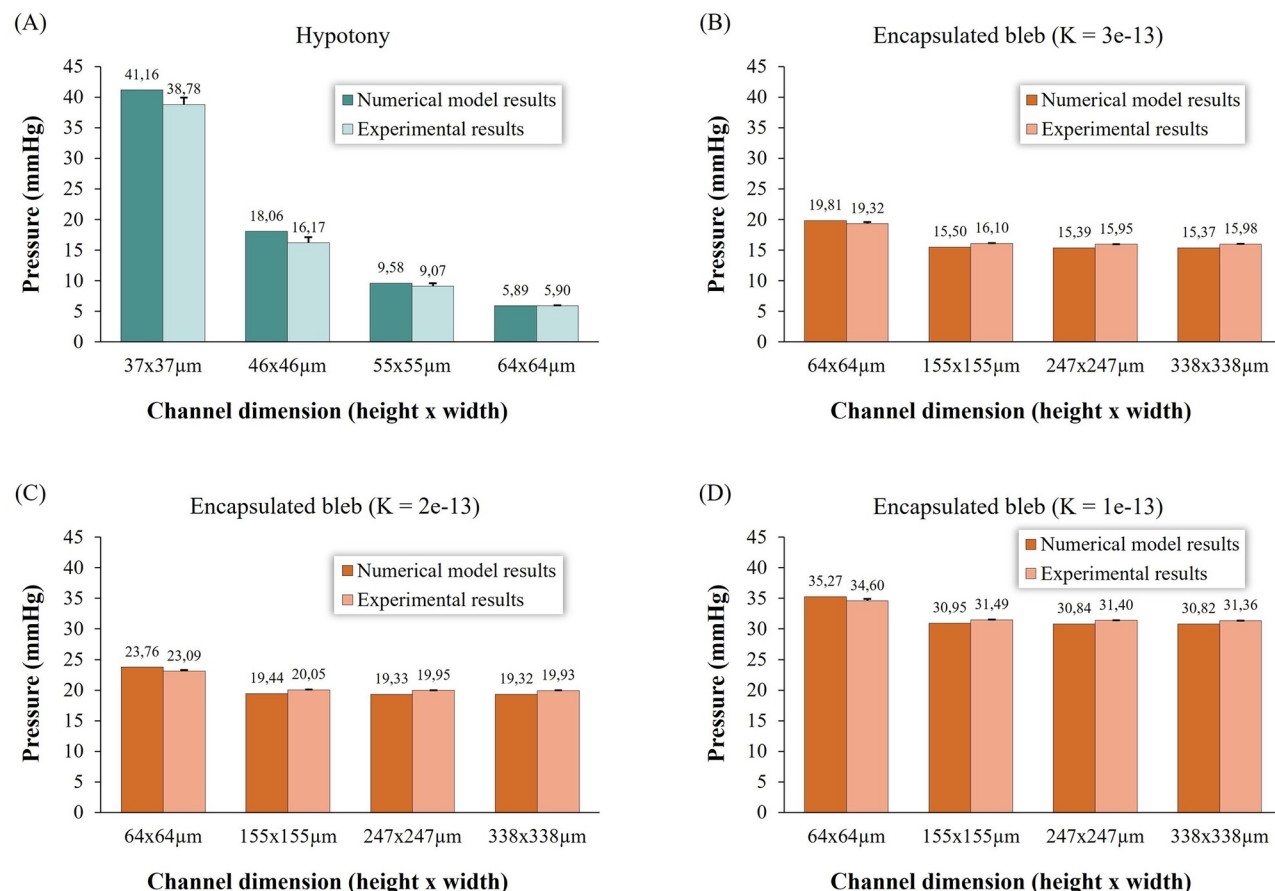

**Fig 11. Calculated IOP vs. experimental IOP.** Comparison between the IOPs calculated with the model and the IOPs measured in the microfluidic experiments in case of **(A)** hypotony and **(B)**, **(C)** and **(D)** bleb encapsulation, and for different shunt lumen diameters/channel dimensions.

conductivity of the scar tissue layer becomes (thus, the more severe the fibrotic reaction is), the higher the pressure in the bleb will be and, consequently, the IOP. Our results also indicate that decreasing the hydrodynamic resistance of the glaucoma drainage device by enlarging its lumen diameter will not decrease the IOP sufficiently to reach acceptable values when the bleb is already encapsulated—only a 3-mmHg pressure drop is achieved. In contrast, in a hypotony situation, increasing the hydrodynamic resistance of the implant by decreasing its lumen diameter does effectively increase the IOP from the hypotonic values to a healthy IOP range. These results were confirmed and validated by performing microfluidic experiments using microdevices containing channels with distinct hydrodynamic resistances. In conclusion, our model, as well as the numerical results, may provide guidelines to help designing future (patient-specific) glaucoma implants with adjustable hydrodynamic resistances, where the outflow of aqueous humor through such devices could be fine-tuned postoperatively according to the IOP measured in the patient's eyes. This way, common postoperative complications, such as hypotony, can be avoided.

## Supporting information

**S1 Data.**
(XLSX)

## Author Contributions

**Conceptualization:** Inês C. F. Pereira, Jaap M. J. den Toonder.

**Data curation:** Inês C. F. Pereira.

**Formal analysis:** Inês C. F. Pereira.

**Funding acquisition:** Henny J. M. Beckers.

**Investigation:** Inês C. F. Pereira.

**Methodology:** Inês C. F. Pereira.

**Project administration:** Henny J. M. Beckers, Jaap M. J. den Toonder.

**Resources:** Henny J. M. Beckers, Jaap M. J. den Toonder.

**Software:** Inês C. F. Pereira.

**Supervision:** Hans M. Wyss, Jaap M. J. den Toonder.

**Validation:** Inês C. F. Pereira.

**Visualization:** Inês C. F. Pereira.

**Writing – original draft:** Inês C. F. Pereira.

**Writing – review & editing:** Inês C. F. Pereira, Hans M. Wyss, Leonard Pinchuk, Henny J. M. Beckers, Jaap M. J. den Toonder.

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
