## [Decision Letter · Decision Letter 0]

8 Jul 2022

PONE-D-22-04970

A computational model for designing intraocular pressure-regulating glaucoma implants

PLOS ONE

Dear Dr. Tonder,

Thank you for submitting your manuscript to PLOS ONE. After careful consideration, we feel that it has merit but does not fully meet PLOS ONE’s publication criteria as it currently stands. Therefore, we invite you to submit a revised version of the manuscript that addresses the points raised during the review process.

We look forward to receiving your revised manuscript.

Kind regards,

Adélia Sequeira, Ph.D

Academic Editor

PLOS ONE

Additional Editor Comments:

Below please find reviewers comments for the above-mentioned article. As you will see, major and minor revisions are recommended. The editor would like to invite a resubmission of the manuscript, which must address the issues raised by both referees. A revised version should be resubmitted within due date, to be sent for further review.

Reviewers' comments:

Reviewer's Responses to Questions

**Comments to the Author**

1. Is the manuscript technically sound, and do the data support the conclusions?

Reviewer #1: Yes

Reviewer #2: Yes

2. Has the statistical analysis been performed appropriately and rigorously? 

Reviewer #1: N/A

Reviewer #2: N/A

3. Have the authors made all data underlying the findings in their manuscript fully available?

Reviewer #1: Yes

Reviewer #2: Yes

4. Is the manuscript presented in an intelligible fashion and written in standard English?

Reviewer #1: Yes

Reviewer #2: Yes

5. Review Comments to the Author

Reviewer #1: This paper aims to provide a computational model to simulate the IOP in the glaucoma scenario in the presence of a drainage implant. In this case, the IOP is determined by the aqueous humour dynamics that is produced in the ciliary body, flows to the anterior chamber and is removed through an implanted device to a filtering bleb - a space bounded by the sclera and conjunctiva/ Tenon’s capsule. It is clear that the physiological properties of the tissues that define the filtering bleb have an important role on the IOP, namely the existence of scar layer.

Based on the reference [15], the authors present a mathematical model defined by a system of partial differential equations:

- Navier-Stokes equations for the aqueous humour in the shunt domain;

- Darcy’s law coupled with incompressibility equation in the bleb, scar layer and subconjunctival tissue.

The system of equations is completed with:

-no-flux conditions at external boundaries (surface in contact with the sclera, conjunctival barrier);

-continuity of the flux and pressure at the interface between different domains (bleb and subconjunctiva, bleb and scar tissue, scar tissue and subconjunctiva).

Numerical results illustrating the IOP behavior in different scenarios are included. The numerical results are compared with experimental results based in microfluidic experiments.

There are some questions that need to be analyzed before the acceptance of the paper:

1-What are the main differences between the mathematical model presented here and the one included in the reference [15]?

2-How are coupled the equations for the fluid in the shunt domain with Darcy’s equation in the bleb?

3-The interface and the boundary conditions should be written mathematically.

4-The main objective of the paper is to present a computational model. It should be pointed out that the computational model is not the mathematical model presented in the paper. In fact, to define a computational model is necessary to apply numerical methods to the partial differential equations. These methods allow the replacement of the continuous problem by a set of discrete equations that can be used to develop a software package.

In the paper we do not have any reference to the computational methods used to approximate the system of partial differential equations and their interface and boundary conditions.

5-In line 58, the authors state that the fluid in the bleb can be absorbed by the subconjunctival or episcleral vasculature. This fact means that the fluid can be absorbed by the sclera and go to these blood vessels? Why this fact is not included in the mathematical model?

6-The bleb is a space filled with aqueous humour. The authors use Darcy’s law in description of the aqueous humour dynamics. This fact means that the bleb is seen as a porous medium. This is the right approach? Why the Navier-Stokes equations are not used?

Reviewer #2: The manuscript is sound and contains the data used in the simulations and numerical experiments.

Glaucoma is the damage of the optical nerve caused by a high intraocular pressure (IOP). The different types of treatment (drug therapy, laser, surgical) aim at lowering IOP. The paper addresses the problem of tuning the properties of a glaucoma drainage device surgically implanted- namely the hydraulic resistance-in order to prevent ulterior hypotony (IOP excessive lowering) or the increase of IOP due to bleb scarring.

The manuscript presents two complementary approaches: a mathematical model of aqueous humor drainage, based on a Coupled Navier Stokes+Darcy model, and a laboratorial model based on a prototype where microfluidic experiments are performed.

The manuscript is interesting and it provides insights into the design of future glaucoma implants. However, we consider that a certain number of questions should be clarified:

1) The authors model the flow in the shunt with Navier-Stokes equation and the permeation through the bleb and the conjunctiva with Darcy equation. Nonetheless, the physiological properties of the bleb suggest that modelling the flow there with Navier-Stokes equation is more appropriate and could lead to results that are more rigorous. Arguments explaining the rationale under the authors modelling choice in the bleb should be presented.

2) Regarding numerical simulations with Comsol Multiphysics, no details are discussed namely the need for local mesh refinements (interface shunt/bleb/scar tissue?).

3) The model accounts for shunt geometry before application, but not to its deformation (curvature) after insertion.

4) As stated in the paper the aqueous humor production varies with the circadian rhythm. The authors present their simulations with an aqueous humor production Qin=2,5 �l/min. Wouldn't using a weighted average of AH production (during 24 hours) have led to a more realistic model?

5) The authors mention that they essentially follow the model in Gardiner [15]. The originality of the contribution of the present manuscript should be clearly stated.

6. PLOS authors have the option to publish the peer review history of their article (what does this mean?). If published, this will include your full peer review and any attached files.

Reviewer #1: **Yes: **J.A. Ferreira

Reviewer #2: No

---

## [Author Response · Author response to Decision Letter 0]

23 Jul 2022

Our response to the reviewer comments is provided in a separate document.

---

## [Decision Letter · Decision Letter 1]

12 Aug 2022

A model for designing intraocular pressure-regulating glaucoma implants

PONE-D-22-04970R1

Dear Dr. Toonder,

We’re pleased to inform you that your manuscript has been judged scientifically suitable for publication and will be formally accepted for publication once it meets all outstanding technical requirements.

Kind regards,

Adélia Sequeira, Ph.D

Academic Editor

PLOS ONE

Additional Editor Comments (optional):

The revised version of the manuscript has been substantially improved and can now be accepted for publication.

Reviewers' comments:

Reviewer's Responses to Questions

**Comments to the Author**

1. If the authors have adequately addressed your comments raised in a previous round of review and you feel that this manuscript is now acceptable for publication, you may indicate that here to bypass the “Comments to the Author” section, enter your conflict of interest statement in the “Confidential to Editor” section, and submit your "Accept" recommendation.

Reviewer #1: All comments have been addressed

Reviewer #2: All comments have been addressed

2. Is the manuscript technically sound, and do the data support the conclusions?

Reviewer #1: Yes

Reviewer #2: Yes

3. Has the statistical analysis been performed appropriately and rigorously? 

Reviewer #1: N/A

Reviewer #2: N/A

4. Have the authors made all data underlying the findings in their manuscript fully available?

Reviewer #1: Yes

Reviewer #2: Yes

5. Is the manuscript presented in an intelligible fashion and written in standard English?

Reviewer #1: Yes

Reviewer #2: Yes

6. Review Comments to the Author

Reviewer #1: The paper can now be accepted for publication. The authors addressed to all my questions. In my opinion, the quality of the paper was improved.

Reviewer #2: The paper is interesting. The results are sound and can be of great help for medical doctors and pharmaceuticals developing ophthalmic devices. All our comments have been addressed in the present version.

7. PLOS authors have the option to publish the peer review history of their article (what does this mean?). If published, this will include your full peer review and any attached files.

Reviewer #1: No

Reviewer #2: No

---

## [Editor Report · Acceptance letter]

25 Aug 2022

PONE-D-22-04970R1 

A model for designing intraocular pressure-regulating glaucoma implants 

Dear Dr. den Toonder:

I'm pleased to inform you that your manuscript has been deemed suitable for publication in PLOS ONE. Congratulations! Your manuscript is now with our production department. 

Kind regards, 

on behalf of

Dr. Adélia Sequeira 

Academic Editor

PLOS ONE